# Estimation of Spatial and Seasonal Variability of Soil Erosion in a Cold Arid River Basin in Hindu Kush Mountainous Region Using Remote Sensing

**Ziauddin Safari [1], Sayed Tamim Rahimi [1], Kamal Ahmed [2], Ahmad Sharafati [3,\*], Ghaith Falah Ziarh [1], Shamsuddin Shahid [1], Tarmizi Ismail [1], Nadhir Al-Ansari [4,\*], Eun-Sung Chung [5] and Xiaojun Wang [6,7]**

[1] School of Civil Engineering, Faculty of Engineering, Universiti Teknologi Malaysia, Johor Bahru 81310, Malaysia; zia.safari2011@gmail.com (Z.S.); tamim.rahimi@graduate.utm.my (S.T.R.); eng.ghaith.ziarh@gmail.com (G.F.Z.); sshahid@utm.my (S.S.); tarmiziismail@utm.my (T.I.)

[2] Department of Water Resource Management, Lasbela University of Agriculture, Water and Marine Sciences, Uthal, Lasbela, Balochistan 90150, Pakistan; kamalahmed.est@luawms.edu.pk

[3] Department of Civil Engineering, Science and Research Branch, Islamic Azad University, Tehran, Iran

[4] Civil, Environmental and Natural Resources Engineering, Lulea University of Technology, 97187 Lulea, Sweden

[5] Department of Civil Engineering, Seoul National University of Science and Technology, Seoul, Korea; eschung@seoultech.ac.kr

[6] State Key Laboratory of Hydrology-Water Resources and Hydraulic Engineering, Nanjing Hydraulic Research Institute, Nanjing 210029, China; xjwang@nhri.cn

[7] Research Center for Climate Change, Ministry of Water Resources, Nanjing 210029, China

\* Correspondence: asharafati@srbiau.ac.ir (A.S.); nadhir.alansari@ltu.se (N.A.-A.)

**Abstract:** An approach is proposed in the present study to estimate the soil erosion in data-scarce Kokcha subbasin in Afghanistan. The Revised Universal Soil Loss Equation (RUSLE) model is used to estimate soil erosion. The satellite-based data are used to obtain the RUSLE factors. The results show that the slight (71.34%) and moderate (25.46%) erosion are dominated in the basin. In contrast, the high erosion (0.01%) is insignificant in the study area. The highest amount of erosion is observed in Rangeland (52.2%) followed by rainfed agriculture (15.1%) and barren land (9.8%) while a little or no erosion is found in areas with fruit trees, forest and shrubs, and irrigated agriculture land. The highest soil erosion was observed in summer (June–August) due to snow melting from high mountains. The spatial distribution of soil erosion revealed higher risk in foothills and degraded lands. It is expected that the methodology presented in this study for estimation of spatial and seasonal variability soil erosion in a remote mountainous river basin can be replicated in other similar regions for management of soil, agriculture, and water resources.

**Keywords:** Fluvisol; RUSLE; data scarcity; remote sensing; Afghanistan

## 1. Introduction

Soil erosion harms agriculture productivity due to the reduction of soil fertility [1–4] Besides, it affects water quality by reducing the level of dissolved oxygen [5–8]. Soil erosion is caused by different drivers including water and air [9]. A large portion of global land (1094 Mha) is altered by soil erosion due to hydrological processes, of which a significant portion (224 Mha) cannot be restored [10]. The global average of soil erosion by either precipitation or floods is 20–30 Gtyr-1 [11].

Soil erosion rates are higher in arid and semi-arid regions due to high sensitivity of such region to climate and land-use changes [12–17]. The soil erosion rates are also reported to be higher in Asian countries like China, India, and Afghanistan [18–20]. Therefore, the arid region of Asia is considered as highly susceptible to soil erosion. Soil erosion is a slow natural geologic phenomenon, but it costs billions of dollars for agricultural

losses [21–23]. Hence, soil erosion study is a vital concern for the regions with agricultural-based economies.

Afghanistan located between the south and central Asia has a predominantly arid and semi-arid climate. Soil erosion is a challenging issue for agricultural productivity in Afghanistan [20,24]. Shrestha [25] reported that it is complicated to figure out the exact amount of soil erosion in Afghanistan due to a lack of data. The findings from the Global Assessment of Human-induced Soil Degradation (GLASOD) project indicated that a large portion (75%) of the topsoil of Afghanistan is removed due to water and wind. Furthermore, excessive erosions on riverbanks lead to notable social and economic losses. Despite the high importance of the soil erosion problem in Afghanistan, a few studies have been conducted to address the issue over the different regions in Afghanistan [20,26,27].

Sahaar [27] integrated the Revised Universal Soil Loss Equation (RUSLE) with Geographic Information System (GIS) to estimate the annual soil loss of Kabul Rivers. They found that the river was significantly eroded in the rate of 19 ton·acre$^{-1}$·year$^{-1}$. [26] evaluated the annual soil loss of the Lower Harirud watershed in Heart province using RUSLE. The research findings indicated that soil erosion was varied in the range of 0.025 to 778 Mg ha$^{-1}$·y$^{-1}$. [20] assessed the volume of eroded soil from Gardez basin in Paktya province, Afghanistan using the Universal Soil Loss Equation (USLE) integrated with GIS. Results showed that annual soil loss was found to be in the range of 0 to more than 100 ton·ha$^{-1}$·y$^{-1}$. The findings reported in the literature indicated that soil erosion rates are estimated to be significantly high. Thus, further studies should be conducted over the vulnerable basins to provide sustainability in agriculture and environment in the country.

Kokcha subbasin, which is located in Amu Darya River basin in Northeast of Afghanistan, is a data-scarce region with high soil erosion rate. People with agricultural income are dominated in the Kokcha subbasin. Hence, their properties are significantly dependent on the farming industry [28–30]. Rapid deforestation and highly frequent rainfall depth, which change the river flow pattern, provide an increasing trend in soil erosion over the Kokcha subbasin [20,31–33]. Hence, it is essential to estimate the potential amount of soil erosion in the basin and provide the measures required for protecting the vulnerable lands.

A number of approaches have been developed to estimate the soil erosion such as USLE, RUSLE, Agricultural Non-Point Source (AGNPS) model, and Watershed Erosion Prediction Project (WEPP) [34]. Several factors, including precipitation intensity, soil type, physiography, land use, and anthropogenic activities, have significant impacts on soil erosion [35–40]. Hence, soil erosion modelling needs a large number of in situ measurements including hourly rainfall, land use, and digital elevation model data which are not available for the basin [25,31,33]. Satellite-based remote sensing data are widely used to assess the susceptible zones of soil erosion in data-scare regions [34,41,42]. To estimate soil erosion with remote sensing data, the RUSLE method provided better performance [43–45]. The parameters of the RUSLE method can be estimated using the remote sensing data collected from several sources and augmented in geographical information system (GIS). Bahrawi et al. [46] reported that remote sensing data and GIS provided a flexible and powerful tool for assessing soil erosions in data-scarce regions.

This study aims to estimate the soil erosion using the satellite-based remote sensing data in the Kokcha subbasin. In this way, the RUSLE method is employed for estimating soil erosion. The satellite-based products are used to obtain the data required for computing the RUSLE factors due to scarcity of in-situ data in Afghanistan. Cold agricultural regions share a significant portion of global food production [47]. Soil erosion poses a major threat to soil degradation and crop productivity in cold regions. Wu et al. [48] found soil erosion in cold climate is often much higher than that in other climatic regions. Snow melting at the end of winter or early spring causes high overland flow which often exceed the runoff in rainy season of other climatic zones. Reduction of soil infiltration capacity due to freezing enhance surface runoff and thus erosion [49,50]. However, studies related to soil erosion in cold climate, particularly cold mountainous climate region is very limited.

Analysis of seasonal variability and spatial distribution of soil erosion in Kokcha subbasin can contribute valuable knowledge to soil erosion in cold climatic region. The soil erosion maps generated in this study would help to understand the spatial patterns of soil erosion within the Kokcha subbasin to assist stakeholders in taking suitable mitigation measures.

## 2. Study Area and Data Sources

### 2.1. Study Area

Kokcha subbasin, which is located in the north-eastern of Afghanistan, covers an area of 21,811 km² (Figure 1). The basin is situated from 35.40° to 37.40° N and 68.90° to 71.60° E. Additionally, the altitude of the basin is varied from 370 m above sea level in the northwestern part to 6721 m in the southern mountains. The percentage of the basin area above 2000, 3000, and 4000 m are respectively, 75%, 50%, 30%. The area receives an annual average rainfall of 470 mm [30]. The snow covers in the basin are approximately found above 4000 m. The mountainous areas in the basin are furnished with low-density trees. The basin is characterized by cold semi-arid and warm continental climate. Rainfall events mainly occur in spring (March–May) and autumn (October–April) seasons while winter (December–February) is usually cold with heavy snowfalls. Several small glaciers are observed in the Kokcha Basin. The glaciers and snow are the primary water resources to recharge the river in the summer (June–August) [51]. Soil in most parts of the basin can be categorized as Fluvisol based on Food and Agriculture Organization (FAO) classification.

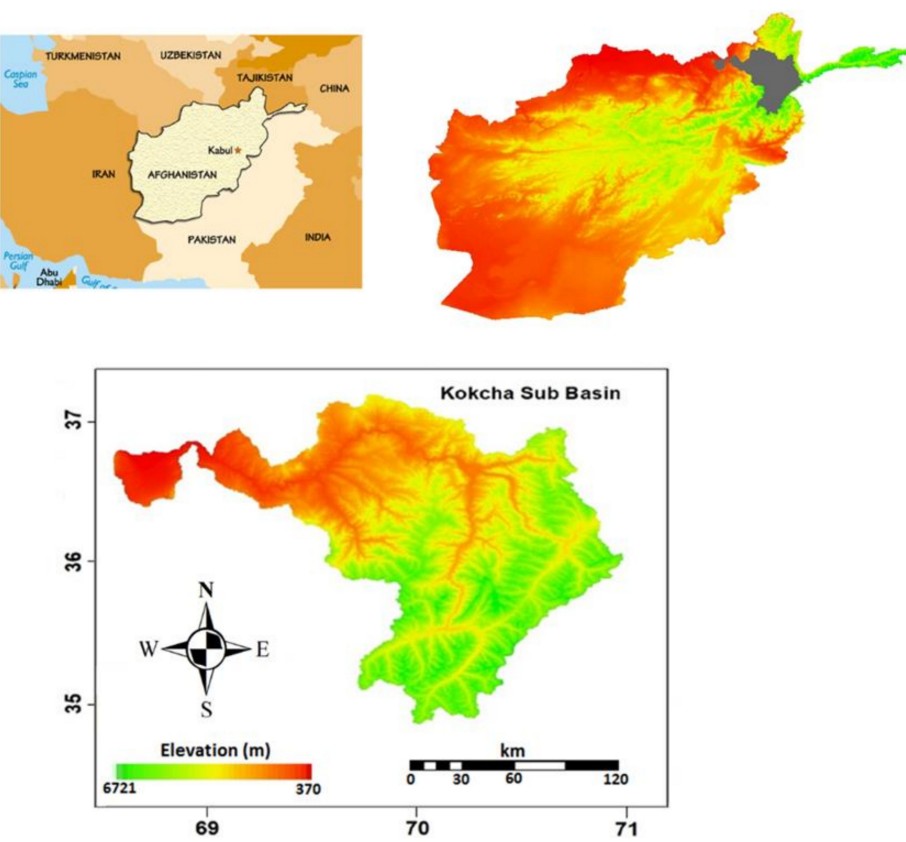

**Figure 1.** Geographical location and topography of the Kokcha subbasin.

### 2.2. Data Sources

Most of the hydrological and climatological stations in Afghanistan are destroyed due to four decades of internal war [30,31]. Hence, adequate observation data are rarely found for a national scale study. Thus, the satellite-based data, which are freely available for large areas, are preferred. The digital elevation model (DEM) of the basin with 30 m resolution is obtained from Advanced Space Thermal Emission Radiometer (ASTER). The Global

Precipitation Measurement (GPM) product (GPM_3IMERGHH, half-hourly, $0.1° \times 0.1°$) obtained by National Aeronautics and Space Administration (NASA) is used to provide the rainfall erosivity factor (R) data. The Normalized Difference Vegetation Index (NDVI) provided by Moderate Resolution Imaging Spectroradiometer (MODIS) is employed to extract the cover and management factor (C). Slope length and steepness (LS) factors are derived from DEM of Afghanistan, while soil erodibility factor (K) is obtained from the Soil Types Map from ISRIC World Soil Information. All the data are obtained at a spatial resolution of $0.1° \times 0.1°$.

## 3. Methodology

The RUSLE method has been widely used for predicting the rate of inter-rill and rill erosions[52]. The RUSLE integrated with GIS is used in the present study to predict soil erosion in the Kokcha subbasin. The RUSLE factors are calculated and mapped using remote sensing data and GIS techniques. The RUSLE method is defined as follows:

$$A = R \times K \times LS \times C \times P \tag{1}$$

where $A$ is the potential annual soil erosion (t ha$^{-1}$ year$^{-1}$), $R$ is rainfall/runoff erosivity factor, $K$ is soil erodibility factor, $LS$ is the slope length/steep factor, $C$ is vegetation cover factor, and $P$ is support practice factor.

### 3.1. Rainfall Erosivity Factor

The rainfall erosivity factor (R) reflects the effect of rainfall intensity on soil erosion [38,53]. A long period rainfall time series is required to compute the accurate $R$ factor; however, that is not possible for many regions like Kokcha subbasin. To solve this limitation, the half-hourly GPM product is used to provide the rainfall data required in the present study.

The formula proposed by Wischmeier and Smith [53] is used to compute the $R$ factor as the following equation:

$$R = \frac{\sum_{i=1}^{m}(916 + 331 \ \log I_i)I30_i}{N} \tag{2}$$

where $R$ denotes the mean annual rainfall erosivity (100 foot-ton in acre$^{-1}$ h$^{-1}$ year$^{-1}$), m indicates the number of storms in the N-year period, $I_i$ is the mean intensity of the $i$-th storm (in h$^{-1}$), $I30_i$ is maximum 30-min intensity of the $i$-th storm (in h$^{-1}$).

### 3.2. Soil Erodibility Factor

The soil erodibility factor (K) quantifies the soil erosion caused by runoff [54]. Several parameters, namely soil texture, organic content, permeability and soil type, have significant impact on $K$ factor [55]. Herein, the Erosion Productivity Impact Calculator (EPIC) is used to compute the K factor (100 acre)$^{-1}$ ft$^{-1}$ Mg$^{-1}$ in$^{-1}$ as follows [56]:

$$K = \left[0.2 + 0.3 \exp\left(0.0256 S_a(1 - \frac{s_i}{100})\right)\right]\left(\frac{s_i}{cl + s_i}\right)^{0.3}$$
$$\left(1 - \frac{0.25 \ C}{C + \exp(3.72 - 2.95 \ C)}\right)\left(1 - \frac{0.7 \ SN}{SN + \exp(-5.51 + 22.9SN)}\right)0.1317 \tag{3}$$

where $Cl$, $Sa$ and $Si$ are the fraction (%) of clay, sand and silt respectively; and $C$ is the soil organic carbon content (%).

### 3.3. Slope Length and Steepness Factor

The impacts of the slope length and slope gradient on soil erosion is quantified using the slope $LS$ factor [54]. The accuracy of the $LS$ value depends on the spatial resolution of the DEM, availability of computing resources, and the scale of the study area. DEMs

with spatial resolution coarser than 100 m do not accurately capture the flow network of a catchment [44]. Hence, the *LS* factor is computed based on a DEM with a resolution of 30 m using the following formula:

$$LS = \left(\frac{Q_a M}{22.13}\right)^y \times \left(0.065 + 0.045S + 0.0065S^2\right) \tag{4}$$

where $Q_a$ is flow accumulation grid; $S$ is grid slope in percentage; $M$ is grid size (x × y), and $y$ is a dimensionless exponent in the range of 0.2 to 0.5.

### 3.4. Land Cover Management Factor

The Land Cover Management Factor (C) is used to express the effect of cropping and management practices on soil erosion [52]. The factor is most sensitive to human activities [57]. The Normalized Difference Vegetation Index (*NDVI*) is used to calculate the *C* factor in intraseasonal and interannual timescales [58]. The C factor can be computed as the following expression [59,60]:

$$C = \exp\left[-\alpha\left(\frac{NDVI}{\beta - NDVI}\right)\right] \tag{5}$$

where $\alpha$ and $\beta$ are the dimensionless parameters. van der Knijff et al. [60] proposed the best values of $\alpha$ and $\beta$ were 2 and 1, respectively.

### 3.5. Support Practice Factor

Conservation or Support Practice (P) factor is quantified the practices that affect the runoff rate and thus, the soil erosion rate [52]. It denotes the ratio of soil erosion provided after a practice (i.e., contour farming, cross-slope cultivation, and strip cropping) to the regular farming slope [52]. The P-factor is considered 1 due to a lack of proper data of conservation and support activities in the basin.

### 3.6. Validation of Soil Erosion Model

The suspended sediment loads estimated by Hydrology Department of Ministry of Energy and Water of Afghanistan at Khawajaghar station, which is located at the basin outlet, are used as an observed dataset for the validation of RUSLE estimated soil erosion. The monthly erosion pattern of observed and modelled soil erosion is compared to show the ability of RULSE model to estimate the seasonal variability of observed soil erosion.

## 4. Result and Discussion

### 4.1. Spatial Pattern of the RUSLE Factors

To estimate the soil erosion in the Kokcha subbasin, the maps of RUSLE factors (e.g., R, K, LS, and C) are prepared using the ArcMap software (Figure 2). The map of R factor is provided based on the mean annual rainfall data (Figure 2a). It is characterized based on the five classes in the range of 174 to 317 (MJ mm ha$^{-1}$·h$^{-1}$·y$^{-1}$). The highest erosivity is observed in the southern of the basin while the northwestern region exposes the lowest values. The central parts of the basin represent the rainfall erosivity from 244 to 266 MJ mm ha$^{-1}$·h$^{-1}$·y$^{-1}$. A large area of the basin from north to south attains the erosivity in the range of 266 to 285 MJ mm ha$^{-1}$·h$^{-1}$·y$^{-1}$.

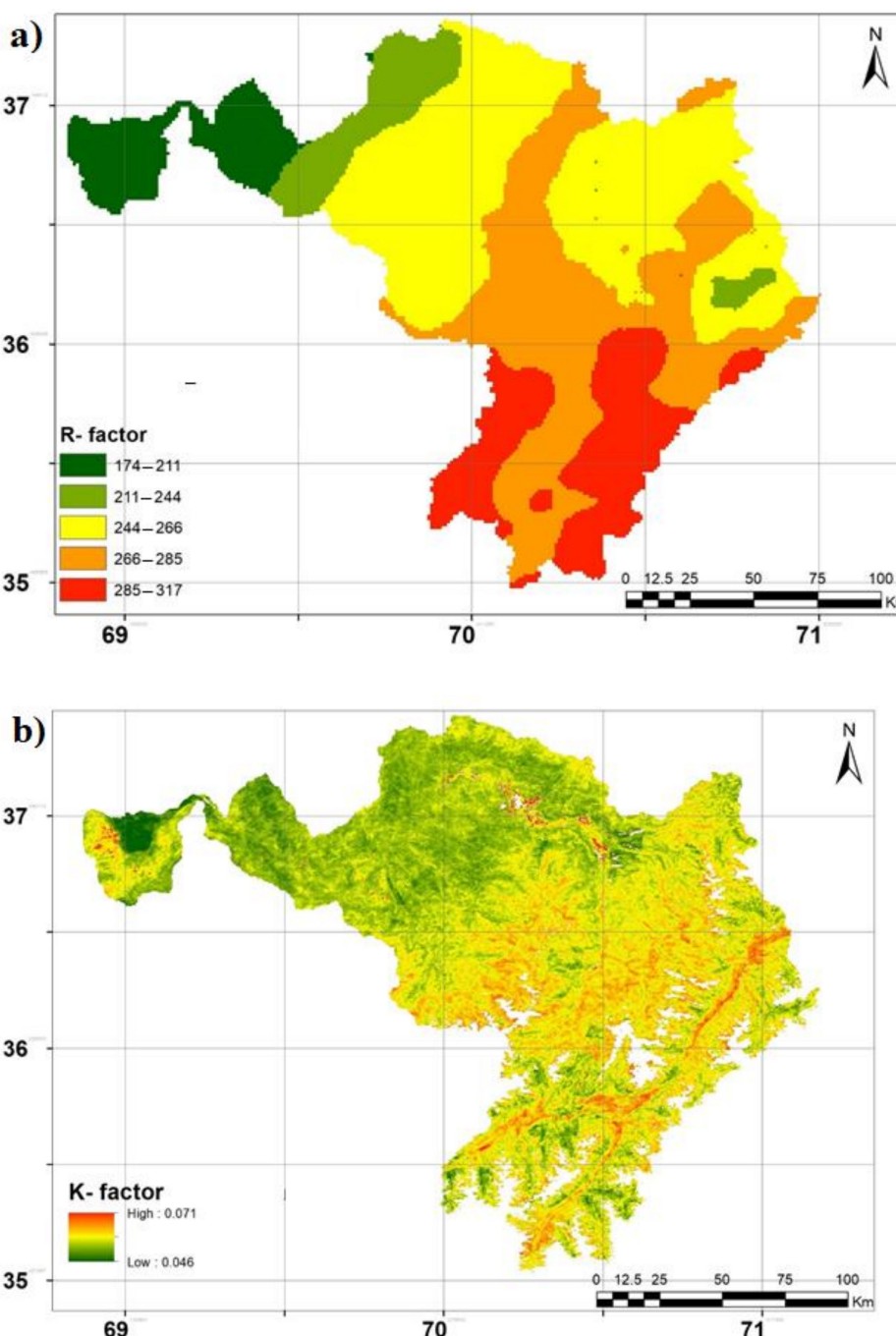

**Figure 2.** *Cont*.

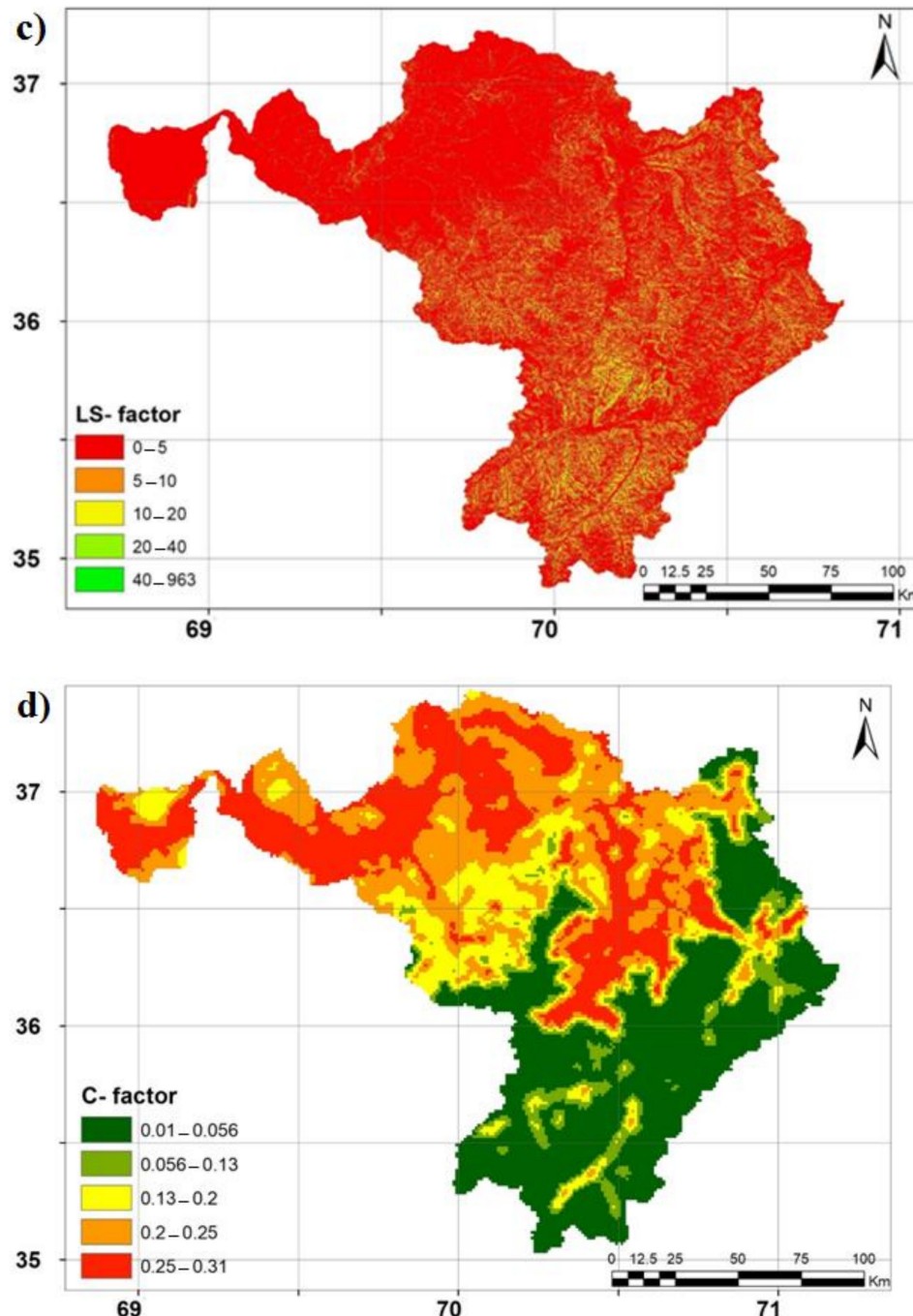

**Figure 2.** The maps of the RUSLE factors over the Kokcha subbasin. (**a**) Rainfall erosivity (R) factor, (**b**) soil erodibility (K) factor, (**c**) slope length and steepness (LS) factor, and (**d**) land cover management (C) factor.

Figure 2b represents that the K values are in the range of 0.046 to 0.071 t ha h ha$^{-1}$ MJ$^{-1}$·mm$^{-1}$, where the highest soil erodibility factor values are found in the southern area of the basin. The LS map of the Kokcha subbasin, which is provided by a 30 m DEM, is shown in Figure 2c. From the figure, it is clear that the LS values varied from 0 to 963. Besides, the highest LS values mostly lie in the eastern and southern parts of the basin.

Figure 2d exposes that the C factor in the Kokcha subbasin are characterized based on the five classes, including (0.01–0.056), (0.056–0.13), (0.13–0.20), (0.20–0.25), and (0.25–0.31). However, the basin can be approximately divided to lower- and upper-half regions based on the two dominated classes (e.g., 0.01–0.056 and 0.25–0.31). The highest and the lowest C values, respectively, are seen in the south and north of the basin.

## 4.2. Spatial Pattern of Soil Erosion

Having provided the RUSLE factors maps, the annual soil erosion pattern in Kokcha can be obtained using Equation (1) through the raster calculator of spatial analyst tool in ArcMap software (Figure 3). Jenk's optimum classification algorithm (JOCA) [61] is used for classification of estimated soil erosion values to prepare the map. Based on JOCA, the soil erosion (t ha$^{-1}$·y$^{-1}$) in basin was classified into five classes including slight [0–10), moderate [10–50), high [50–150), severe [150–500), and very severe (>500). From Figure 3, it is evident that slight erosion is significantly observed over the study area, while moderate erosion is mainly found in the center area of the basin. The other erosion classes, including high, severe, and very severe, are rarely observed over the Kokcha subbasin.

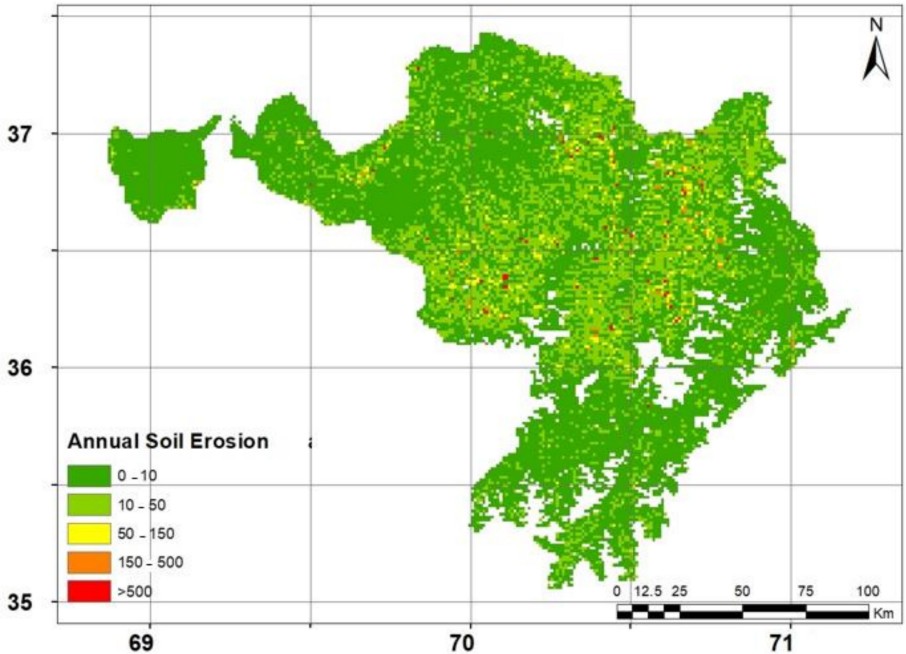

**Figure 3.** Annual soil erosion map of Kokcha subbasin.

Table 1 demonstrates the percentage of soil erosion in the classes defined in the present study. As shown in the table, it is clear that the slight erosion (0–10 t·ha$^{-1}$·y$^{-1}$) is found in 71.34%. The moderate erosion (10–50 t·ha$^{-1}$·y$^{-1}$), which has the second rank in vastity, covers 25.46% of the basin. The other erosion classes include 3.2% of the Kokcha subbasin in the mountains/foothills and degraded lands. The high slope steepness, very poor vegetation and no conservation practices are the most prominent causes of soil erosion over the Kokcha subbasin.

**Table 1.** Description of the soil erosion rate classes in Kokcha subbasin.

| Soil Loss (t·ha$^{-1}$·y$^{-1}$) | Risk Categories | Area (ha) | Area (%) |
|---|---|---|---|
| 0–10 | Slight | 1,555,970.20 | 71.34 |
| 10–50 | Moderate | 555,345.51 | 25.46 |
| 50–150 | High | 53,096.23 | 2.43 |
| 150–500 | Severe | 16,521.36 | 0.76 |
| >500 | Very Severe | 227.88 | 0.01 |
| Total | | 2,181,161.17 | 100.00 |

There is no universal rule for classification of soil erosion values to prepare erosion map. Therefore, different classification schemes were adopted in different studies. Classification of soil erosion class is subjective which can significantly affect the areas covered by different soil erosion zones. JOCA can be used to avoid subjectivity in classification of

soil erosion. JOCA classifies values in such a way that the variability of the values within a class is least but among the class is maximum. This ensures the best arrangement of erosion values in different classes. JOCA has been widely used for classification of environmental data for which any generalized classification scheme is not available [62].

*4.3. Validation of the RUSLE Model*

The available observed soil erosion data (Table 2) is used to validate the results. Table 2 shows that 26 suspended sediment samples for different months are available for the period of 2014–2017.

**Table 2.** Observed data of Khawajaghar sediment gauging station.

| Date | G .H (m) | Mean Discharge m³/s | Suspended Sediment | |
| --- | --- | --- | --- | --- |
| | | | Conc ppm | Discharge (t/d) |
| 4 May 2014 | 2.72 | 149.1 | 7572.22 | 97,547.21 |
| 17 August 2014 | 3.06 | 223.7 | 79.99 | 1546.10 |
| 24 December 2014 | 2.08 | 48.06 | 156.98 | 651.82 |
| 27 January 2015 | 2.05 | 52.36 | 103.30 | 467.32 |
| 28 February 2015 | 2.16 | 76.42 | 347.00 | 2291.13 |
| 13 April 2015 | 2.36 | 124.2 | 1818.00 | 19,508.74 |
| 4 May 2015 | 2.42 | 150.4 | 3338.00 | 43,375.84 |
| 16 June 2015 | 3.48 | 451.85 | 2016.00 | 78,704.32 |
| 5 July 2015 | 4.1 | 796 | 5008.00 | 344,422.20 |
| 13 August 2015 | 2.63 | 412.3 | 2680.00 | 95,468.89 |
| 13 September 2015 | 2.63 | 186.2 | 186.90 | 3006.79 |
| 2 November 2015 | 2.23 | 128.3 | 86.27 | 956.31 |
| 26 January 2016 | 2.15 | 91.29 | 115.40 | 910.21 |
| 16 March 2016 | 2.02 | 79.68 | 270.90 | 1864.97 |
| 12 April 2016 | 2.44 | 180.9 | 1158.00 | 18,099.26 |
| 18 May 2016 | 2.95 | 259.2 | 4385.00 | 98,201.55 |
| 15 June 2016 | 3.8 | 517.8 | 6520.00 | 291,691.24 |
| 27 July 2016 | 3.3 | 326.9 | 2058.00 | 58,126.48 |
| 5 September 2016 | 2.72 | 164.4 | 124.00 | 1761.32 |
| 9 October 2016 | 2.42 | 127 | 659.00 | 7231.08 |
| 6 December 2016 | 2.14 | 95.7 | 302.00 | 2497.08 |
| 28 February 2017 | 2.68 | 77.23 | 679.54 | 4534.33 |
| 1 June 2017 | 2.68 | 166.25 | 8020.10 | 115,200.77 |
| 30 July 2017 | 3.58 | 329.78 | 2734.08 | 77,902.12 |
| 12 September 2017 | 2.99 | 185.31 | 1065.87 | 17,065.46 |
| 22 October 2017 | 2.6 | 108 | 359.45 | 3354.05 |
| Daily mean | | | 43,611.3 | |
| Annual mean | | | 17,847,711.6 | |

Figure 4 represents the monthly pattern of soil erosion over the Kokcha subbasin. Figure shows the highest erosion rate in summer (June, July and August) due to snow melting from high mountains. This finding has good agreement with those obtained from the Khawajaghar station where the maximum sediment loads are observed in June and July months. The total annual soil loss estimated for the study area (22,116,974 t) is slightly higher than the mean annual sediment load observed at Khawajaghar station (17,847,711 t). The RUSLE model provides either the underestimated soil erosions due to ignoring the gully erosion, channel erosion, bank erosion, and mass wasting events such as landslides or the overestimated soil losses due to missing the deposition and sediment routing [44]. Hence, the difference between the measured and estimated erosion is acceptable.

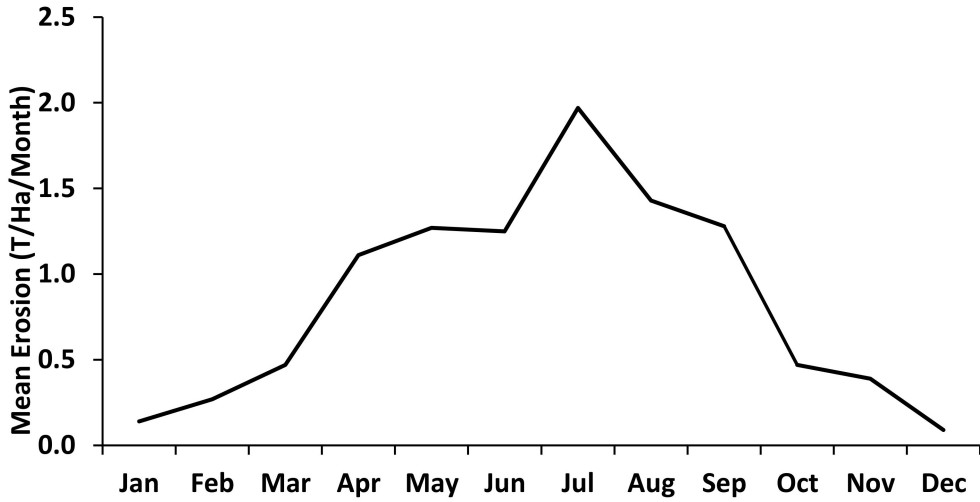

**Figure 4.** Monthly soil erosion over of the study area.

*4.4. Assessing the Soil Erosion over the Different Land Use and Land Cover*

To assess the soil erosion over different land use classes of the Kokcha subbasin, the land use/land cover map of the study area is intersected with the soil erosion map. Table 3 demonstrates the percentage of soil erosion for different land use types. It is clear from the table that Rangeland (52.2%) followed by rainfed agriculture (15.1%) and barren land (9.8%) provides the highest percentage of erosion. The erosion in snow-covered area was found 3.4%. The erosion of snow-covered area might be due to an increase in temperature over Afghanistan [31]. Additionally, slight erosion is found in regions with fruit trees, forest and shrubs, and irrigated agriculture land.

**Table 3.** Mean annual soil loss rate in different LULC.

| LULC | Area (ha) | Area (%) | Percentage of Erosion (%) |
|---|---|---|---|
| Barren land | 212,751.5 | 9.75 | 9.8 |
| Barren land/Rangeland | 103,046.9 | 4.72 | 4.7 |
| Built-up | 7005.24 | 0.32 | 0.3 |
| Forest and shrubs | 17,885.57 | 0.82 | 0.8 |
| Forest and shrubs/Rangeland | 623.43 | 0.03 | 0.0 |
| Fruit Trees | 3451.92 | 0.16 | 0.2 |
| Fruit Trees/Irrigated agriculture land | 1053.97 | 0.05 | 0.0 |
| Irrigated agriculture land | 60,604.52 | 2.78 | 2.8 |
| Irrigated agriculture land/ Fruit Trees | 3887.35 | 0.18 | 0.2 |
| Rainfed agriculture land | 329,655.3 | 15.11 | 15.1 |
| Rainfed agriculture land/Forest and shrubs | 949.65 | 0.04 | 0.0 |
| Rainfed agriculture land/Rangeland | 16,992.78 | 0.78 | 0.8 |
| Rangeland | 1,139,352 | 52.24 | 52.2 |
| Rangeland/Barren land | 164,595 | 7.55 | 7.5 |
| Rangeland/Rainfed agriculture land | 210,07.04 | 0.96 | 1.0 |
| Snow-covered | 74,825.5 | 3.43 | 3.4 |
| Water Bodies | 23,473.47 | 1.08 | 1.1 |

Table 4 illustrates the soil erosion classes (slight, moderate, high, severe, very severe) of different land use existed in the Kokcha subbasin. As presented in the table, it is obvious that the highest erosion rates with different classes are found in the rangeland which followed by rainfed agriculture land and barren land. Due to the importance of rainfed agriculture land, the government should take urgent actions for soil conservation, sustainable land use practices for controlling soil erosion risks.

**Table 4.** Soil erosion of Kokcha subbasin area (ha) based on land use/land cover.

| LULC | Area (ha) | Soil Erosion (t ha$^{-1}$ y$^{-1}$) | | | | |
|---|---|---|---|---|---|---|
| | | Slight 0–10 | Moderate 10–50 | High 50–100 | Severe 100–500 | Very Severe >500 |
| Barren land | 212,751.54 | 151,770.10 | 54,168.68 | 5179.03 | 1611.50 | 22.23 |
| Barren land/Rangeland | 103,046.92 | 73,510.35 | 26,236.78 | 2508.48 | 780.54 | 10.77 |
| Built-up | 7005.24 | 4997.31 | 1783.61 | 170.53 | 53.06 | 0.73 |
| Forest and shrubs | 17,885.57 | 12,758.99 | 4553.85 | 435.39 | 135.48 | 1.87 |
| Forest and shrubs/Rangeland | 623.43 | 444.73 | 158.73 | 15.18 | 4.72 | 0.07 |
| Fruit Trees | 3451.92 | 2462.49 | 878.89 | 84.03 | 26.15 | 0.36 |
| Fruit Trees/Irrigated agriculture land | 1053.97 | 751.87 | 268.35 | 25.66 | 7.98 | 0.11 |
| Irrigated agriculture land | 60,604.52 | 43,233.31 | 15,430.52 | 1475.30 | 459.05 | 6.33 |
| Irrigated agriculture land/Fruit Trees | 3887.35 | 2773.11 | 989.76 | 94.63 | 29.45 | 0.41 |
| Rainfed agriculture land | 329,655.34 | 235,165.51 | 83,933.56 | 8024.83 | 2497.00 | 34.44 |
| Rainfed agriculture land/Forest and shrubs | 949.65 | 677.45 | 241.79 | 23.12 | 7.19 | 0.10 |
| Rainfed agriculture land/Rangeland | 16,992.78 | 12,122.10 | 4326.53 | 413.66 | 128.71 | 1.78 |
| Rangeland | 1,139,351.95 | 812,777.02 | 290,090.4 | 27,735.4 | 8630.10 | 119.04 |
| Rangeland/Barren land | 164,594.97 | 117,416.76 | 41,907.53 | 4006.75 | 1246.74 | 17.20 |
| Rangeland/Rainfed agriculture land | 21,007.04 | 14,985.75 | 5348.60 | 511.38 | 159.12 | 2.19 |
| Snow | 74,825.50 | 53,378.10 | 19,051.32 | 1821.48 | 566.77 | 7.82 |
| Water Bodies | 23,473.47 | 16,745.22 | 5976.58 | 571.42 | 177.80 | 2.45 |
| Total Area (ha) | 2,181,161.17 | 1,555,970 | 555,346 | 53,096 | 16,521 | 228 |
| Percentage | | 71.34 | 25.46 | 2.43 | 0.76 | 0.01 |

## 5. Conclusions

This study proposes an approach to estimate the mean annual soil erosion over the data-scare Kokcha subbasin in Afghanistan. In this way, the RUSLE model, coupled with GIS, is used to estimate soil erosion using the satellite-based data. The following conclusions are obtained from the present study. First, the main fraction (71.34%) of the basin is characterized based on the slight erosion class, while the small area (0.01%) of the study area is induced by very severe soil erosion risk. Second, the central parts of the Kokcha subbasin provide the highest erosion rate. Third, the soil erosion risk is found to be higher in regions with steep slopes, very poor vegetation, and inadequate conservation practices. Fourth, the severe and very severe soil erosion risks are mainly found in mountains/foothills and degraded lands. Fifth, the results of the study have adequate consistency with those obtained from measured data in Khawajaghar station. Sixth, the rangeland has the highest percentage of soil erosion followed by rainfed agriculture land and barren land. Comparison with observed data showed that RUSLE model is able to simulate the seasonal variability of observed soil erosion in the study area. However, a slight overestimation by RUSLE is noticed which may be due to ignoring the deposition and sediment routing. High erosion from agricultural land can significantly reduce soil quality and agricultural sustainability in the region. Different cost-effective adaptation measures including natural confinements of agricultural land, conservation tillage and plant residue management should be adopted for mitigation of soil erosion from agricultural land. Satellite-based precipitation and NDVI data used in this study have considerable uncertainty in estimation. In future, precipitation and NDVI data from different satellite products can be used to assess uncertainty in simulated soil erosion.

**Author Contributions:** Conceptualization: S.S. and E.-S.C.; methodology: S.T.R. and K.A.; software: Z.S., K.A., and G.F.Z.; validation: N.A.-A., X.W., and S.S.; formal analysis: T.I.; investigation: Z.S.; resources: X.W.; data curation: S.T.R.; writing—original draft preparation: Z.S. and G.F.Z.; writing—review and editing: A.S., E.-S.C., and S.S.; visualization: A.S.; supervision: S.S., N.A.-A.; project administration: T.I.; funding acquisition: N.A.-A. All authors have read and agreed to the published version of the manuscript.

**Funding:** This research received no external funding.

**Institutional Review Board Statement:** Not applicable.

**Informed Consent Statement:** Not applicable.

**Data Availability Statement:** The data that support the findings of this study are available from the corresponding author upon justifiable request.

**Acknowledgments:** Authors are grateful to the Ministry of Education, Government of Afghanistan for providing financial support to complete this study. Authors also acknowledge the support provided by Universiti Teknologi Malaysia.

**Conflicts of Interest:** The authors declare no conflict of interest.

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
