# Peer review of "Estimation of Spatial and Seasonal Variability of Soil Erosion in a Cold Arid River Basin in Hindu Kush Mountainous Region Using Remote Sensing"

_sustainability, doi:10.3390/su13031549_

Round 1

Reviewer 1 Report

This study aims to estimate the spatial and seasonal variability of soil erosion using remote sensing. I think that this study is interesting. Using modeling to estimate and predict soil erosion in countries that data availability is critical, these kinds of studies might be beneficial.

This manuscript is publishable after revising. The introduction is structured suitably. The purpose is clear too.  

Line 113, P 3: What is the average precipitation for this basin? If you do not have data, you can use available global maps, e.g., NOAA.

Line 234, P 10: I think that the validation method (besides the existing issues) should be explained in Methods.

The discussion of the manuscript is a little week. In Section 5, the authors only report results.

Author Response

This study aims to estimate the spatial and seasonal variability of soil erosion using remote sensing. I think that this study is interesting. Using modeling to estimate and predict soil erosion in countries that data availability is critical, these kinds of studies might be beneficial.

Response: Thank you very much for finding our paper interesting.

This manuscript is publishable after revising. The introduction is structured suitably. The purpose is clear too. 

Response: Thnak you very much

Line 113, P 3: What is the average precipitation for this basin? If you do not have data, you can use available global maps, e.g., NOAA.

Response: The area receives an annual average rainfall of 470 mm. This has been mentioned with proper reference.

Line 234, P 10: I think that the validation method (besides the existing issues) should be explained in Methods.

Response: Thank you very much for your comment. A new section (Section 3.6) is added in method section to discuss how the model output was validated.

The discussion of the manuscript is a little week. In Section 5, the authors only report results.

Response: Thank you for your comment. New texts are added to state the implication of the study. Besides, limitation of the study and recommendation for future work are also provided.

Reviewer 2 Report

Authors assess soil erosion in a study area in Afghanistan by using the well-known Revised Universal Soil Loss Equation (RUSLE). Giving that the area is data poor, RUSLE factors are derived through remote sensing.

I think the paper is suitable for SUSTAINABILITY journal. The general presentation is good and the methodology is well explained. However, the main points to be clarified are the following:

  1. I have not got clear an important question: why the study is important, i.e., what is its contribution to Science? I think the authors should make clearer these points.
  2. Which is the criterion used to reclassify soil erosion map into 5 classes? How have been the limits of each class defined (L220-221)? You should also explain the rationality behind your choice giving that it strongly affects the extent of each soil erosion class. In the discussion session, some considerations about the influence of the reclassification method on the resulting map are necessary.
  3. I suggest to invert the colour scale of fig. 3 so that the red colour is assigned to areas with the highest erosion rates.

Author Response

Authors assess soil erosion in a study area in Afghanistan by using the well-known Revised Universal Soil Loss Equation (RUSLE). Giving that the area is data poor, RUSLE factors are derived through remote sensing.

I think the paper is suitable for SUSTAINABILITY journal. The general presentation is good and the methodology is well explained. However, the main points to be clarified are the following:

  1. I have not got clear an important question: why the study is important, i.e., what is its contribution to Science? I think the authors should make clearer these points.

Response: Thank you for your comment. New texts have been added in the last paragraph of Introduction section to discuss the importance of the study and contribution to knowledge.

  1. Which is the criterion used to reclassify soil erosion map into 5 classes? How have been the limits of each class defined (L220-221)? You should also explain the rationality behind your choice giving that it strongly affects the extent of each soil erosion class. In the discussion session, some considerations about the influence of the reclassification method on the resulting map are necessary.

Response: The method used for classification is mentioned in the paper. Besides, influence of classification is discussed in the end of section 4.2. Following texts have been added for this purpose:

Jenk’s optimum classification algorithm (JOCA) (Jenks, 1967) is used for classification of estimated soil erosion values to prepare the map.

….

There is no universal rule for classification of soil erosion values to prepare erosion map. Therefore, different classification schemes were adopted in different studies. Classification of soil erosion class is subjective which can significantly affect the areas covered by different soil erosion zones. JOCA can be used to avoid subjectivity in classification of soil erosion. JOCA classifies values in such a way that the variability of the values within a class is least but among the class is maximum. This ensures the best arrangement of erosion values in different classes. JOCA has been widely used for classification of environmental data for which any generalized classification scheme is not available (Ahmed et al., 2015).

  1. I suggest to invert the colour scale of fig. 3 so that the red colour is assigned to areas with the highest erosion rates.

Response: Thank you for your comment. We revised the color scheme. We also believe that the figure now provides better information after reversing the color.

Other minor revisions suggested in attached file by the reviewer is also made in the revised manuscript.
